# The Effects of Intermittent Hypoxic Training on Anaerobic and Aerobic Power in Boxers

**DOI:** 10.3390/ijerph17249361

**Published:** 2020-12-14

**Authors:** Tadeusz Ambroży, Marcin Maciejczyk, Andrzej T. Klimek, Szczepan Wiecha, Arkadiusz Stanula, Piotr Snopkowski, Tomasz Pałka, Janusz Jaworski, Dorota Ambroży, Łukasz Rydzik, Wojciech Cynarski

**Affiliations:** 1Institute of Sports Sciences, University of Physical Education, 31-571 Kraków, Poland; tadek@ambrozy.pl (T.A.); janusz.jaworski@awf.krakow.pl (J.J.); dorota.ambrozy@awf.krakow.pl (D.A.); 2Department of Physiology and Biochemistry, Faculty of Physical Education and Sport, University of Physical Education in Kraków, 31-571 Kraków, Poland; marcin.maciejczyk@awf.krakow.pl (M.M.); andrzej.klimek@awf.krakow.pl (A.T.K.); tomasz.palka@awf.krakow.pl (T.P.); 3Department of Rehabilitation, Faculty of Physical Education and Sport in Biała Podlaska, Józef Piłsudski University of Physical Education, 00-809 Warsaw, Poland; szczepan.wiecha@awf-bp.edu.pl; 4Institute of Sport Science, The Jerzy Kukuczka Academy of Physical Education, Mikołowska 72A, 40-065 Katowice, Poland; a.stanula@awf.katowice.pl; 5Doctoral School, University of Physical Education in Kraków, 31-571 Kraków, Poland; boxinghome76@gmail.com; 6Institute of Physical Culture Studies, College of Medical Sciences, University of Rzeszów, 35-310 Rzeszów, Poland; ela_cyn@wp.pl

**Keywords:** aerobic capacity, anaerobic capacity, boxing, normobaric hypoxia

## Abstract

Background: The aim of the study was to evaluate the effects of intermittent hypoxic training (IHT) on anaerobic and aerobic fitness in elite, national boxers. Methods: The study was conducted over a period of 6 weeks. It comprised 30 national championship boxers, divided into 2 groups: the experimental and control. Both groups performed the same boxing training twice a day (morning and afternoon training). In the afternoon, the experimental group performed training under normobaric conditions in a hypoxic chamber (IHT), while the control group undertook exercise in standard normoxic conditions. In both groups, before and after the 6-week programme, basic anthropometric indices as well as anaerobic (Wingate Test) and aerobic (graded test) fitness were assessed. Results: There was a significant increase in anaerobic peak power (988.2 vs. 1011.8 W), mean anaerobic power (741.1 vs. 764.8 W), total work (22.84 vs. 22.39 kJ), and a decrease in fatigue index (20.33 vs. 18.6 W·s^−1^) as well as time to peak power (5.01 vs. 4.72 s). Such changes were not observed in the control group. In both groups, no significant changes in endurance performance were noted after the training session – peak oxygen uptake did not significantly vary after IHT. Conclusions: Our results have practical application for coaches, as the IHT seems to be effective in improving anaerobic performance among boxers.

## 1. Introduction

In recent years, scientific interest in boxing has increased significantly. The data published are related to the structure of training in individual mesocycles, predicting sport results as well as physiological profile of competitors [1,2,3]. In some publications, results of studies are presented concerning only the motor skills of boxers [4,5]. Only a handful of publications have been devoted to assessing indices of anaerobic capacity and muscle damage in boxers [6,7].

Currently, constant monitoring of exercise intensity and evaluation of physiological responses to boxing training can be carried out by measuring heart rate with sports testers, while the level of lactate concentration can be measured on the earlobes. Training volume is defined as an effective duration of exercise. Training loads are computed as the product of the training volume and its intensity using the notation of loads in energetic and informative zones [8].

In boxing, it is necessary to conduct training that provides comprehensive motor preparation, including that focused on speed, strength, power, coordination, and physical fitness of athletes, which is one of the conditions for success in this sport. Unfortunately, there is a lack of specific training guidelines, which makes it difficult to determine whether the applied training methods are optimal to maximise exercise capacity [9]. Boxing requires not only a combination of technical, tactical, mental and physical skills, but also high aerobic fitness, which, along with strength and speed, is certainly one of the most important features that should be considered when planning a boxer’s physical conditioning programme [9,10]. The dynamics of movement around a ring, frequent changes in effort intensity, delivering blows and defense responses are primarily based on anaerobic metabolism, hence the need to develop them in the course of the training cycle. Therefore, physical training in boxing should be aimed at increasing both the aerobic and anaerobic fitness of a boxer.

A boxing competition, as in other combat sports, requires both aerobic and anaerobic metabolism in the effort during a fight. Competitors usually attack with maximal strength, which requires anaerobic metabolism. Aerobic metabolism usually occur when a competitor relaxes after an attack or during breaks between rounds. The frequency of both types of processes also depends on the number of rounds in a bout, which is why amateur competing more frequently requires aerobic energy pathway in comparison to a professional boxing competition with a greater number of rounds [11].

For this reason, new methods or combinations of training methods are sought to maximise exercise capacity. One such method is residing in the lowlands and training at an appropriate altitude (Live Low/Train High [LL/TH]). Recently, coaches, mainly of endurance disciplines, but also of strength-power disciplines, include hypoxic training in their programmes. In studies on hypoxic training, improvement in exercise capacity of athletes is suggested [12]. One of the such methods is intermittent hypoxic training (IHT). In accordance with this method, athletes undergo 1–2 h of training in a hypoxic chamber before and after being in normoxic conditions. 

The physiological mechanism of the body’s response to hypoxia is used to increase both aerobic and anaerobic fitness. In previous studies on IHT, improvement has been demonstrated in the ability to perform long-term physical efforts of sub-maximal intensity; an increase in maximal oxygen uptake (VO_2_max) and intensity at metabolic thresholds—particularly the second ventilatory threshold (VT_2_), is extremely important in competitive sports [13,14]. However, in other studies, increases in VO_2_max as a result of IHT were not noted, which may be explained by the insufficient exercise duration in hypoxic conditions and the overall workload [15,16]. Research results also indicate that training in hypoxic conditions increases anaerobic power, which characterises the ability to perform short-term physical exercise at maximal and supramaximal intensities [17,18,19,20]. However, some authors suggested that this type of training does not significantly affect anaerobic capacity or explosive strength of the lower limbs or maximal running speed [21,22].

Due to ambiguous opinions regarding the effectiveness of IHT [13,16], the aim of this study was to assess the impact of IHT on anaerobic and aerobic fitness in elite, national boxers. We adopted the hypothesis that including IHT in standard boxing training, will significantly improve aerobic and anaerobic fitness in boxers, further allowing to develop new training methods in this sport with the use of hypoxic training.

## 2. Materials and Methods

### 2.1. Experimental Approach to the Problem

The study was designed as pretest/posttest evaluation. The research was experimental, conducted over a period of 6 weeks. The study comprised 30 men: Polish elite, male boxers, who were randomly divided into 2 groups of 15: the experimental (IHT) group and control (normoxia) group, ensuring the same number of competitors from similar weight categories (light, medium and heavy) be included in both groups, which further performed the same standard physical training 5 days a week. In the morning, a 60-min technical boxing training session was performed, including exercises conducted with low to medium intensity (up to 50% of maximum load, up to HR_max_), and in the afternoon, a 60-min boxing training session. Details of the training are presented in Table 1 and Table 2. The only difference in training was that boxers in the experimental group performed the afternoon training in normobaric hypoxic conditions in a hypoxic chamber, while competitors from the control group remained in standard normoxic conditions. The athletes trained in Kraków (Poland), at an altitude of about 230 m (754.5 ft) above sea level. During the experiment, the subjects were asked to maintain their usual diet, which was verified using entries in food diaries. IHT was implemented in a normobarcic hypoxic chamber at a simulated altitude of 4000 m (FiO2 = 12.9%). During the tests, the chamber temperature was 21–22 °C with an air humidity of 40–45%. All boxers fully completed all components of the study.

Prior to participation in the tests, the competitors were informed about the research procedures, which were in accordance with the ethical principles of the Declaration of Helsinki WMADH (2000). Obtaining the competitors’ written consent was the condition for their participation in the project. The research was approved by the Bioethics Committee at the Regional Medical Chamber (No. 42/KBL/OIL/2015).

### 2.2. Participants

The studied athletes (elite, national, male boxers) had a minimum of ten years’ training experience, and were winners of medals at national and international competitions. The mean age of boxers was 24.2 ± 3 yrs (IHT) and 23.5 ± 3 yrs (normoxia).

### 2.3. Somatic Measurements and Stress Tests

Basic anthropometric indices and the level of aerobic and anaerobic fitness were assessed in both groups before and after completing the 6-week training cycle.

Body mass and fat mass were determined with a body composition analyser (Tanita, MC 718, Japan) using the method of electrical bioimpedance. Body height was measured via a stadiometer. Body mass index (BMI) was calculated for each of the subjects. The measurements were carried out in the morning under euhydration (proper body hydration conditions). For 24 h before any measurements, participants did not engage in prolonged physical exercise or sauna use. Their feet were clean and degreased. All measurements were carried out at similar ambient temperatures.

The exercise tests took place on 2 consecutive days: the anaerobic power test (Wingate test) was performed on the first day, and the next day, endurance was assessed by measuring peak oxygen uptake (VO_2_peak) and by determining the level of the second ventilatory threshold. Exercise tests were carried out in normoxic conditions (FiO2 20.93%) at a room temperature of 21 ± 0.5 °C and relative humidity of 40 ± 5%. During this time, the men could consume fluids without any restrictions, and before going to bed and in the morning after waking up, they were advised to drink a total of 1000 mL of isotonic fluids. Twenty-four hours before the stress tests, the competitors were not allowed to consume alcohol, coffee or other stimulants. All of the tests were carried out in the early morning hours, following a light breakfast.

To determine the level of maximal anaerobic power, the 30-s Wingate test was applied (Bar-Or, 1987), which was performed on a Cyclus 2 cycloergometer (RBM elektronik-automation GmbH, Leipzig, Germany), with a load totalling 7.5% of the subject’s body mass. The main test was preceded by a 5-min warm-up at 100 W. During the warm-up, the subjects performed two 5-s maximal accelerations in the 2nd and 4th minutes. Two minutes after the warm-up, the subjects performed the Wingate test. The athletes began the effort in a static position, their task being to develop maximal pedaling cadence on the cycloergometer as quickly as possible and then maintain it for as long as possible. During the test, the following indices were measured: total work performed (TW), peak anaerobic power (PP), average power after 30 s of the test (MP), time to attain peak power (tPP), and power decrease index (FI). Immediately before and during the 3rd minute, as well as 20 min after the completion of exercise, 20 µL of blood were collected from the fingertip for determination of lactate concentration (LA) using the Super GL2 analyser (Müller Gerätebau GmbH, Freital, Germany).

In order to assess the level of aerobic fitness – peak oxygen uptake (VO_2_peak) and the second ventilatory threshold—a running test of progressive intensity was performed on the mechanical treadmill (h/p/cosmos, Nussdorf—Traunstein, Germany). The test began with a 2-min recording of ventilatory markers at rest, the subjects in a standing position. For the first 4 min of the test, the participants ran at a speed of 8 km·h^−1^, afterwards, the running speed was increased by 1 km·h^−1^ every 2 min. The effort was continued until volitional exhaustion, which was manifested in the inability to keep on running at the determined speed. During the test, the levels of cardiopulmonary indices were recorded based on the “breath-by-breath” method using an ergospirometer (Cosmed, Rome, Italy). The following indices were analysed: pulmonary ventilation (V_E_), oxygen uptake (VO_2_), carbon dioxide production (VCO_2_), respiratory-exchange-ratio (RER), expiratory carbon dioxide concentration (%F_E_CO_2_), ventilatory equivalent ratio for oxygen and carbon dioxide (V_E_/VCO_2_), and heart rate (HR). Data were averaged every 30 s. The highest registered value of oxygen uptake was considered as peak oxygen uptake. The second ventilatory threshold (VT_2_) was determined based on the dynamics of changes in respiratory indices. It was assumed that VT_2_ was reached after the following criteria were met: (1) a decrease in %F_E_CO_2_ after reaching maximal level; (2) a rapid nonlinear increase in V_E_ (second deflection); (3) the V_E_/VCO_2_ ratio reached a minimum and began to increase; (4) a nonlinear increase in VCO_2_ (second deflection) [23].

Immediately before and during the 3rd and 20th minutes after completion of the progressive test, blood was collected from the fingertip for determination of lactate concentration using a Super GL2 analyser (Dr. Müller Gerätebau GmbH, Freital, Germany).

### 2.4. Statistical Methods

The test results were statistically analysed by determining mean values and standard deviations. Normality and homogeneity of variance were confirmed using the Shapiro-Wilks and Levene’s tests. Two-way analysis of variance with repeated measures was used to investigate the main effects and interaction between the group (hypoxia vs. normoxia) and time factors (pre-training vs. post-training). In the case of significant effects regarding the main factors, the significance of differences between specific averages was checked using the Bonferroni test (post-hoc analysis). Moreover, in order to quantify the size of differences in the data, effect size (ES: Cohen’s *d*), as the difference in group means divided by the standard deviation of the pooled data, was calculated and classified as trivial (≤0.19), small (0.20–0.49), moderate (0.50–0.79) or large (≥0.80). For all analyses, the level of *p* ≤ 0.05 was selected to indicate statistical significance. All calculations were performed using STATISTICA ver. 13.3 (TIBCO Software Inc., Palo Alto, CA, USA).

## 3. Results

The physical characteristics of boxers were similar in both groups and are presented in Table 3. There was no significant improvement in somatic variables after the training in either of the groups (Table 3).

Analysis of variance showed significant improvement in absolute peak power (f = 11.225, *p* = 0.007), absolute (f = 12.346, *p* = 0.003) and relative mean power (f = 13.829, *p* = 0.002), fatigue index (f = 7.316, *p* = 0.002), total work (f = 4.548, *p* = 0.049), time to attain peak power (f = 4.535, *p* = 0.048) maximal pulmonary ventilation (f = 6.681, *p* = 0.02), maximal running speed (f = 8.529, *p* = 0.01), and speed at VT_2_ (f = 5.841, *p* = 0.028) in boxers after the training. Post-hoc analysis indicated that significant changes of these variables occurred only in the hypoxia group (Table 4). 

In the hypoxia group, we noted a significantly higher level of the following indices in the Wingate test performed after IHT: peak power (988.2 vs. 1011.8 W; *p* = 0.001, d = 0.192 – small improvement), mean power (741.1 vs. 764.8 W; *p* = 0.001, d = 0.275—small improvement), and total work (22.39 vs. 22.84 kJ; *p* = 0.046, d = 0.18—small improvement). Significant decreases in fatigue index (20.33 vs. 18.6 W·s^−1^; *p* = 0.013, d = 0.594—medium-to-large improvement), and time to attain peak power (5.01 vs. 4.72 s; *p* = 0.038, d = 0.362—small-to-medium improvement) were noted. Similar changes were not found in the group of boxers training in normoxic conditions (Table 4).

In neither of the examined groups were any significant changes noted in endurance abilities, i.e., peak oxygen uptake and VT_2_ threshold level, were noted under the influence of the training (Table 5). However, in the group of boxers training under hypoxic conditions, there was a significant increase in maximal pulmonary ventilation (161.9 vs. 156.5 L·min^−1^; *p* = 0.004, *d* = 0.276—small-to-medium improvement), maximal running speed (15.29 vs. 14.61 km·h^−1^; *p* = 0.002, *d* = 0.531—medium improvement), and running speed at VT_2_ level (13.14 vs. 12.69 km·h^−1^; *p* = 0.019, *d* = 0.454—medium improvement) (Table 5).

## 4. Discussion

Our study involved high-performance athletes: the group that is most interested in using IHT for sports training. To the best of our knowledge, this was the first type of study including boxers. The aim of our research was to assess the impact of specific training in normobaric hypoxic conditions on aerobic and anaerobic physical fitness of competitive boxers. In the study, a significant increase in speed and strength (anaerobic metabolism), as compared to the control group, was shown. Both phosphagen (PP and tPP increase) and glycolytic systems (MP increase and FI decrease) slightly improved. Despite the lack of changes in the value of VO_2_peak and the intensity at VT_2_, training in hypoxic conditions resulted in improvement regarding maximal pulmonary ventilation and an increase in tolerance for higher exercise loads, both during maximal efforts and at the VT_2_. In boxers, the intensity at VT_2_ in the pretest was high (90%VO_2_peak). This was probably due to the high initial level that we did not notice any effects of IHT on VT_2_.

The ‘living low–training high’ method of training in the form of IHT was used in this study by introducing training in a normobaric hypoxic chamber. However, LL-TH altitude training can be carried out using 2 methods: intermittent hypoxic exposure (IHE), when athletes remain under normobaric hypoxia (at a simulated altitude of 3000–6000 m above sea level), alternating with normoxic conditions for 1.5 to 3 h in the post-workout recovery phase, or IHT conditions in which athletes are subjected to normoxic conditions and training units that last from 1 to 2 h in normo- or hypobaric hypoxic conditions. In the former case, being under hypoxic conditions alone causes adaptive reactions that can increase exercise capacity, as seen in the results of this study [24]. Nevertheless, they be insufficient to induce an increase in fitness, as seen in the study by Beidleman et al. [25]. From the point of view of physiological responses to physical exercises in hypoxic conditions, IHT exercise seems to be a preferable form of LL-TH training. Due to the too short exercise time in hypoxia, its effects are not likely to arise from increases in erythrocyte or haemoglobin levels, as suggested by some authors [24,26]. However, they may be caused by other non-hematological reactions, e.g. increased activity of glycolytic enzymes and intensification of buffering reactions in myocytes [13,14,27,28], or lowering the energy cost of effort [29,30]. These types of responses result in an increase in endurance, and thus, in aerobic fitness [13,18,26,29,30,31]. At the same time, they may provide beneficial changes in speed-strength abilities and, furthermore, an increase in the level of anaerobic capacity, as indicated in the results of another study [32]. An increase was found in the exercise capacity among swimmers of 100- and 200-m distances. In a following study [33], favourable impact of IHT on the amount of power generated in the Wingate test was observed. In the present study, it has been confirmed that the use of IHT can be effective in addition to physical training aimed at improving anaerobic power.

Although no significant changes regarding aerobic capacity improvement of boxers were indicated in our study, an increase in running speed at VO_2_peak and at VT_2_ was found. The results of some studies [13,14] demonstrated an increase in maximal oxygen uptake among long-distance runners and cyclists after training in normobaric hypoxic conditions. This was an increase of 3.5–5% and an increase in threshold loads by 4–8%. On the other hand, in other studies, an increase was noted in the duration of progressive exercise until refusal as a result of 6-week IHT in long-distance runners, without showing changes in speed [24]. Therefore, IHT caused an increase in the level of aerobic fitness, which was not confirmed in this study. However, in many previous studies, improvement was not shown in VO_2_max among national elite athletes as a result of IHT [18,23,31,34,35,36,37,38,39]. This may be due to too low exercise loads or insufficient training duration in hypoxic conditions [15,16]. The IHT used in this study did not increase VO_2_peak in the boxers, either, which indicates no changes in the ability to perform aerobic physical exercise. The tested boxers represented a high sports level and from the very beginning of the experiment, they were characterised by a significant level of endurance, which was indicated, among others, by the level of the second ventilatory threshold exceeding approx. 94% HR_max_. The improvement of aerobic fitness is limited in trained individuals, which, in addition to the factors mentioned above, could have been another reason for the lack of improvement in endurance capabilities under the influence of training in hypoxic conditions.

IHT can also be beneficial for anaerobic fitness, which may be used in sports that require high power and speed. Combination of the stress caused by high-power exercise with the stress caused by hypoxic conditions can be a more effective stimulus for exercise adaptation processes than training in conditions of normoxia. In addition, favourable changes in speed-strength abilities during IHT training may result from improvement in phosphocreatine re-synthesis and an increase in phosphofructokinase enzyme activity [28,40,41]. Some authors [17,20] have shown beneficial effects of IHT training on the sprinting ability of cyclists. Improvement in anaerobic capacity of the tested cyclists was probably due to the maximal stress load applied, based on anaerobic energy, in conjunction with hypoxia conditions. Our results confirmed an increase in speed skills obtained after the 6-week IHT training cycle in boxers, although the boxing training that was used here was, of course, different in nature, specific to this discipline. These results may corroborate the hypothesis suggested by some authors [42,43] who indicated short-term, 60–120-min IHT training, combined with resistance exercise as a stimulus for muscle protein synthesis, which, in turn, may be the basis for an increase in speed-strength abilities.

Improvement in anaerobic fitness achieved as a result of hypoxic training was also demonstrated in studies by other authors, who found beneficial effects of a 2-h IHT session on the anaerobic power obtained in the 30-s Wingate test. The trial was conducted for 10 consecutive days at a simulated altitude of 2500 m above sea level [18,19]. Similar results were obtained in our study, in which during the same anaerobic test, a significant increase was noted in peak and average power (small but significant improvement), shortening the time to attain peak power and reducing the decrease in power (fatigue index). Neither in the present study nor in the mentioned above, were any significant changes noted in the control groups training under normoxic conditions. Other authors also showed an increase in the ability to perform repeated sprinting efforts among athletes after applying IHT training in conditions of normobaric hypoxia, which can be partly explained by significant adaptive changes at the molecular level and increased blood flow through active muscles [17,44].

### Limitation of the Study

In our study, we focused on the practical aspect of hypoxic training in boxers. We did not study physiological mechanisms leading to the improvement of the studied indices, i.e., glycotic enzymes, buffering capacity or hypoxia-inducible factor 1. The obtained results indicate a necessity for further research in order to examine physiological mechanisms of the observed changes.

## 5. Conclusions

Our results have vast practical implications for coaches in training of the elite boxers. The intermittent hypoxic training seems to be effective in improving anaerobic performance. The applied intermittent hypoxic training caused a small but significant increase in anaerobic fitness: improvement in peak and mean anaerobic power was observed after the IHT. Despite the lack of significant changes in the most important aerobic fitness indices, i.e., peak oxygen uptake and intensity at the second ventilatory threshold, the applied intermittent hypoxic training resulted in improvement of maximal pulmonary ventilation and an increase in tolerance for higher exercise loads (speed), both during maximal efforts and when exceeding the second ventilatory threshold. Increases of maximal workload (speed) and at VT_2_ after IHT are also beneficial for boxers.

## Figures and Tables

**Table 1 ijerph-17-09361-t001:** Description of exercises performed during the afternoon training session (60 min) in the experimental (hypoxia) and control (normoxia) groups.

Description of exercise	Warm-up, coordination exercises General development exercises, special warm-up exercises specific for the sport, shadow technique. Coordination ladder, tennis-ball exercises.	4. In-pairs boxing technique. 5. Task-technique with partner. 6. Training with punching bags. 7. Technical exercises, series of jabs, hooks, uppercuts and their combination, as well as moving in different directions.	8. Anaerobic endurance training, strength endurance training, power training. 9. Development of special abilities, intensity specific for target competitions. 10. Pace, pace intervals, sprint.	11. Stretching, cooling down the body. 12. Exercises preventing injury after intense work of the body.

**Table 2 ijerph-17-09361-t002:** Course of afternoon training session (60 min) in the experimental (hypoxia) and control (normoxia) groups—description of exercises in Table 1.

Duration	Period in the Course of Testing	Execution of Exercises in the Experimental Programme
First 4 weeks	1st–4th week of testing	Endurance-speed training. Pace intervals of 8 series, 10 s of work at maximal intensity/50 s break for each exercise interspersed with recovery intervals performed at 4 times lower intensity.
Following 2 weeks	5th–6th week of testing	Power training. Power interval of 5 exercises performed at submaximal speed, lasting 20 s, with a 3-min recovery period. During the session, 3 such sets with 10-min recovery (training at low intensity up to 40 percent of maximum load).

**Table 3 ijerph-17-09361-t003:** Participants’ physical characteristics.

Variables	Hypoxia Training				Normoxia Training			
	Before	After	*p*	*d*	Before	After	*p*	*d*
BH (cm)	182.9 ± 5.09	182.9 ± 5.09	-	-	179.7 ± 4.61	179.7 ± 4.61	-	-
BM (kg)	80.9 ± 8.48	80.3 ± 8.35	0.05	0.07	77.7 ± 9.19	78.0 ± 9.16	0.30	0.03
BMI	24.2 ± 1.88	24.0 ± 1.80	0.05	0.11	24.1 ± 2.53	24.2 ± 2.54	0.29	0.04
FAT (%)	15.3 ± 3.84	14.9 ± 3.22	0.47	0.11	14.1 ± 4.49	14.5 ± 4.22	0.36	0.09
FAT (kg)	12.5 ± 4.06	12.1 ± 3.54	0.32	0.11	11.1 ± 4.19	11.5 ± 4.11	0.19	0.10

BH—body height, BM—body mass, BMI—body mass index.

**Table 4 ijerph-17-09361-t004:** Changes in indices of anaerobic capacity under the influence of training in conditions of hypoxia and normoxia.

Variables	Hypoxia Training				Normoxia Training				Interaction Training × Time
	Before	After	*p*	*d*	Before	After	*p*	*d*	
PP [W]	988.2 ± 120.86	1011.8 ± 124.76	0.001 *	0.192	957.8 ± 136.13	955 ± 137.65	0.850	0.020	f = 3.106; *p* = 0.097
MP [W]	741.1 ± 87.26	764.8 ± 84.98	0.001 *	0.275	729.2 ± 78.94	732.3 ± 81.25	0.630	0.039	f = 7.394; *p* = 0.015
PP [W·kg^−1^]	12.21 ± 0.99	12.61 ± 0.81	0.45	0.111	12.28 ± 1.15	12.22 ± 1	0.737	0.056	f = 0.58; *p* = 0.457
MP [W·kg^−1^]	9.16 ± 0.79	9.49 ± 0.83	0.002 *	0.407	9.38 ± 0.65	9.41 ± 0.71	0.631	0.044	f = 9.257; *p* = 0.008
FI [W·s^−1^]	20.33 ± 3.37	18.6 ± 2.37	0.013 *	0.594	16.89 ± 4.79	16.23 ± 4.11	0.371	0.148	f = 1.489; *p* = 0.24
TW [kJ]	22.39 ± 2.63	22.84 ± 2.37	0.046 *	0.180	22.05 ± 2.39	22.19 ± 2.48	0.510	0.057	f = 1.288; *p* = 0.273
tPP [s]	5.01 ± 0.77	4.72 ± 0.83	0.038 *	0.362	4.12 ± 0.36	3.83 ± 0.69	0.340	0.527	f = 0; *p* = 1
LA_3’_ [mmol·L^−1^]	13.1 ± 1.3	12.1 ± 1.62	0.205	0.681	12.1 ± 1.62	10.22 ± 1.58	0.053	1.175	f = 0.633; *p* = 0.438
LA_20’_ [mmol·L^−1^]	10.68 ± 1.98	9.15 ± 2.29	0.252	0.715	9.24 ± 1.93	7.85 ± 1.91	0.162	0.724	f = 0.009; *p* = 0.926
LA_3’-20’_ [mmol·L^−1^]	2.42 ± 1.9	2.95 ± 2.83	0.688	0.220	2.86 ± 1.41	2.37 ± 1.35	0.539	0.355	f = 0.467; *p* = 0.504

PP—peak power; MP—mean power; FI—fatigue index; TW—total work; tPP—time to attain peak power; LA_3_—lactate level 3 min after exercise, LA_20_—lactate level 20 min after exercise, LA_3-20_—difference of metabolising lactate; * significant differences (*p* < 0.05): post-hoc analysis.

**Table 5 ijerph-17-09361-t005:** Changes in indices of aerobic capacity under the influence of training in conditions of hypoxia and normoxia.

Variables	Hypoxia Training				Normoxia Training				Interaction Training × Time
	Before	After	*p*	*d*	Before	After	*p*	*d*	
V_max_ [km·h^−1^]	14.61 ± 1.31	15.29 ± 1.25	0.002 *	0.531	16.26 ± 1.35	16.25 ± 1.11	0.958	0.008	f = 9.009; *p* = 0.008
V_Emax_ [L·min^−1^]	156.5 ± 19.66	161.9 ± 19.53	0.004 *	0.276	165.4 ± 23.8	163.2 ± 20.21	0.712	0.100	f = 1.632; *p* = 0.22
HR_max_ [bpm]	186.6 ± 5.85	187 ± 7.18	0.733	0.061	188.9 ± 6.9	188.3 ± 7.02	0.644	0.086	f = 0.34; *p* = 0.567
VO_2_peak [mL·min^−1^]	4257.8 ± 474.1	4226.1 ± 458.2	0.654	0.068	4438.2 ± 466	4444.7 ± 495.5	0.926	0.014	f = 0.158; *p* = 0.696
VO_2_peak [mL·kg^−1^]	52.74 ± 4.24	52.73 ± 3.44	0.992	0.003	57.36 ± 4.63	57.07 ± 3.13	0.761	0.073	f = 0.051; *p* = 0.824
V_VT2_ [km·h^−1^]	12.69 ± 1.01	13.14 ± 0.97	0.019 *	0.454	13.93 ± 1.06	13.95 ± 0.9	0.833	0.020	f = 4.582; *p* = 0.048
HR_VT2_ [bpm]	176 ± 3.54	175.1 ± 5.25	0.354	0.201	178.3 ± 5.48	178.6 ± 7.2	0.852	0.047	f = 0.58; *p* = 0.459
%HR_max_ at VT_2_	94.38 ± 1.87	93.69 ± 1.98	0.108	0.358	94.44 ± 1.41	94.81 ± 1.21	0.280	0.282	f = 4.55; *p* = 0.049
VO_2_ at VT_2_ [mL·kg^−1^]	48.28 ± 3.32	47.63 ± 3.15	0.241	0.201	50.9 ± 2.88	51.08 ± 2.55	0.688	0.066	f = 1.532; *p* = 0.234
VO_2_ at VT_2_ [mL·min^−1^]	3896.9 ± 386.7	3814.8 ± 383.1	0.082	0.213	3941.5 ± 385.61	3977.3 ± 428.49	0.289	0.088	f = 5.141; *p* = 0.038
%VO_2_peak at VT_2_	91.67 ± 2.84	90.35 ± 2.21	0.128	0.519	88.91 ± 2.77	89.54 ± 1.92	0.482	0.264	f = 2.83; *p* = 0.112
LA_3’_ [mmol·L^−1^]	10.58 ± 1.48	10.58 ± 2.78	0.999	0.000	11.23 ± 1.95	9.68 ± 1.83	0.122	0.820	f = 1.970; *p* = 0.18
LA_20’_ [mmol·L^−1^]	5.6 ± 1.1	4.81 ± 2.02	0.258	0.486	5.86 ± 1.84	4.54 ± 1.66	0.079	0.753	f = 0.328; *p* = 0.575
LA_3’-20’_ [mmol·L^−1^]	4.98 ± 1.66	6.44 ± 2.57	0.127	0.675	5.37 ± 1.35	5.81 ± 2.26	0.523	0.236	f = 0.89; *p* = 0.36

V—velocity; V_E_—pulmonary ventilation; HR—heart rate; VO_2_—oxygen uptake; LA—lactate concentration; max—maximal level; VT_2_—second ventilatory threshold; * significant differences (*p* < 0.05): post-hoc analysis.

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
