# Peer review of "The Effects of Intermittent Hypoxic Training on Anaerobic and Aerobic Power in Boxers"

_ijerph, 2020, doi:10.3390/ijerph17249361_

Round 1

Reviewer 1 Report

Any methods or combinations of training methods resulting in maximizing athletes’ exercise capacity are of high importance for athletes themselves and their coaches. The Authors performed controlled study aimed at evaluating the effects of intermittent hypoxic training (IHT) on the anaerobic and aerobic fitness in national elite boxers. The study is well designed and the manuscript is well written. However I have some minor comments I’d like to express.

  1. How was the FiO2 measured?
  2. I am not sure if the Authors are aware that 0.01 year equals 3.65 days. Taking it into account, did the Authors really counted the age of the participants with such accuracy? They inform, that the mean age of the boxers was e.g. 24.19 ± 3.21 I’m aware that is the result of mathematical operation on the data but the Authors should not provide more precise output than the input data, as a general rule.
  3. What statistical software was used for statistical analysis?
  4. In my opinion, for the clarity, the symbols LA3’; LA20’ and LA3’-20’ might be explained in the table legend, and the reasonable for calculating LA3’-20’ should be provided.
  5. The Authors claim, citing other researchers that the result of their IHT training intervention is rather the result of changes in glycolytic enzymes and intensification of buffering reactions in myocytes or lowering the energy cost of an effort (lines 238-240). However, they did not analyze any of glycolytic enzymes’ activity. Moreover, they also did not analyze e.g. HIF-1 protein expression/concentration after IHT sessions. It would enrich the study to analyze molecular response of boxers’ organisms to and to ensure what kind of response did their training plan trigger.
  6. Author Contributions section needs to be corrected. The Authors should remove the template’s instructions/comment from the text.

Author Response

Dear Reviewer,

Thank you very much for your time and valuable comments, which all have been considered and incorporated. The detailed list of responses is given below. We hope that the modifications and explanation will be acceptable for you.

Yours sincerely,

Rydzik, corresponding author

  1. FiO2 is the concentration of oxygen in the inspiratory air, e.g. the air in the room. It is constantly measured during VO2max test (cardiopulmonary test CPET) by the analyzer (ergospirometer , Cosmed quark CPET, Rome , Italy). Prior to each measurement the device was calibrated using the standard gas according to the manufacturer’s recommendations. Fio2% was automatically recorded by two paramagnetic sensors (one inside the room, another in the ventilation system) working constantly (Paracube sprint, Servomex, UK) and the entire system was computer-controlled.
  2. The age was rounded.
  3. All calculations were performed with STATISTICA ver. 13.3 (TIBCO Software Inc., Palo Alto, CA, USA). – the information was placed in the paper.
  4. Definitions of LA3 and LA20 were placed in the legend. LA3-20 is the difference defining the spped of metabolizing the lactate by the organism. It means also possibility of aerobic processes, because metabolizing the lactate requires such processes and it is done mostly by muscle fibers previously engaged in the production of this metabolite. In our field of study it is assumed to measure this parameter after about 3-minute exercise, because the highest concentration is observed then. See: https://www.ncbi.nlm.nih.gov/pmc/articles/PMC5587270/.
  5. Thank you for the comment. The reviewer is right. This part of discussion is speculative because we did not study glycolytic enzymes, intensification of buffering reactions in myocytes, the energy cost of an effort or HIF-1. We decided to use the conditional tense instead removing it. In our study we focus on the practical aspect of the hypoxic training of the boxers. We did not study the physiological mechanisms that lead to the improvement of the studied indicators. We also added the paragraph titled limitation of the study in wchich we deal with this problem.
  6. The section of author’s contribution was corrected.

Reviewer 2 Report

General comments

The paper aimed to assess the impact of intermittent hypoxic training (IHT) on the level of anaerobic and aerobic fitness in national elite, experienced boxers. The paper is generally well written based on sound literature, the methods are clear, detailed and replicable, the results well presented and discussed with respect to the literature.

When using abbreviations, make sure you explain it first and then use the correct abbreviation throughout the paper.

Introduction

The authors kept in mind that this section is a development of the hypotheses of the study leading to the purpose of the investigation. Although the most relevant literature has been included, I think the section might be improved by adding a small paragraph on the methods used to monitoring trainings (i.e., heart rate, rate of perceived exertion, lactate, etc.) and calculating training loads (i.e., session-RPE, TRIMP, etc.) in boxers.

Line 71-74: Please rephrase. This sentence is a bit confusing.

Materials and Methods

Overall, methods and procedures are clear, detailed and replicable. However, more info about the training intensity and training loads over the period should have been added. Please, provide these details if available. As it is an intervention study of 6-weeks, these information are necessary.

Line 93-95: Please provide more info about the intensity. What do you mean by “up to 50% of maximum load”? Is this relative to heart rate max, VO2 max, 1 repetition max? Please be more precise.

Discussion

I think the results of the study are well discussed with respect to the current literature. However, I think the section sometime lacks for flow. I think at present this should be improved.

Please, provide also the limitations of your study as well as future directions.

Tables

Please check all tables’ format. Tables should be clear, professional and stand on their own. Make sure you check this.

Please provide also tables similar to 1 and 2 for the before noon 60 minutes technical sessions.

Author Response

Dear Reviewer,

Thank you very much for your time and valuable comments, which all have been considered and incorporated. The detailed list of responses is given below. We hope that the modifications and explanation will be acceptable for you.

Yours sincerely,

Rydzik, corresponding author

When using abbreviations, make sure you explain it first and then use the correct abbreviation throughout the paper

A: We checked all abreviations and corrected mistakes.

The authors kept in mind that this section is a development of the hypotheses of the study leading to the purpose of the investigation. Although the most relevant literature has been included, I think the section might be improved by adding a small paragraph on the methods used to monitoring trainings (i.e., heart rate, rate of perceived exertion, lactate, etc.) and calculating training loads (i.e., session-RPE, TRIMP, etc.) in boxers.

A: We added a paragraph about methods of monitoring of the boxing training.

Current constant monitoring of the intensity of exercising and the reaction of the organism during the boxing training can be done by measuring the pulse using sports testers and the level of concentration of lactate measured on the earlobes. The training volume is defined as the effective duration of the exercise. Training loads are computed as the product of the training volume and its intensity using the notation of loads in the energetic and informative zones

Line 71-74: Please rephrase. This sentence is a bit confusing.

A: It was written in more understandable way.

Overall, methods and procedures are clear, detailed and replicable. However, more info about the training intensity and training loads over the period should have been added. Please, provide these details if available. As it is an intervention study of 6-weeks, these information are necessary.

A: The study was controlled by a physician and a coach. The intensity of exercise was between average and maximal. The selection of the intensity was constantly verified with the reaction of the athletes to the training load which was realized according to the authors’ proposition of the training program.  

Line 93-95: Please provide more info about the intensity. What do you mean by “up to 50% of maximum load”? Is this relative to heart rate max, VO2 max, 1 repetition max? Please be more precise.

A: It was up to 50% of maximum pulse rate, it was placed in the paper.

I think the results of the study are well discussed with respect to the current literature. However, I think the section sometime lacks for flow. I think at present this should be improved.

Odp: The discussion was corrected in order to be better readable.

Please, provide also the limitations of your study as well as future directions.

A: We added the section Limitation of the study. In our study we focus on the practical aspect of the hypoxic training of the boxers. We did not study the physiological mechanisms that lead to the improvement of the studied indicators, i.a. glycotic enzymes, buffering capacity or hypoxia-inducible factor 1 (HIF-1). The results of the study indicate the necessity of further research in order to study the physiological mechanism of the observed changes.

Please check all tables’ format. Tables should be clear, professional and stand on their own. Make sure you check this.

Odp: The tables were made according to the editors’ rules. We agree they are a bit unclear, so we used frames in tables 4 and 5. If the editor suggests it.

Please provide also tables similar to 1 and 2 for the before noon 60 minutes technical sessions.

A: Due to editorial limitations we did not place the below table in the paper. It already has 5 tables, but we add this one

Table. Description of exercises performed during the pre-noon training session

Description of the exercise

Warm-up, exercising mobility of body parts

Technical and tactical training, improving technical and tactical elements, individual training (under coach’s supervision), intensity up to 50% HR max

Functional training, multi-joint and multi-surface exercises using boxing techniques, intensity up to 60% HR max

Relaxing and stretching exercises

Relaxing techniques targeted at coping with tension and stress, training with mental coach, individual training

Reviewer 3 Report

Thank you for submitting your article titled "The effects of intermittent hypoxic training on anaerobic and aerobic power in boxers" to IJERPH. Here are some comments and suggestions which I hope will strengthen your manuscript. 

Abstract:

line 22 - would it be correct to say "elite national boxers"?

line 33 - no significant changes "were" noted

line 34 - great practical application but the end of this sentence doesn't really reflect this "great practical application" - I might suggest rewording or adding to the conclusion based on the strength of the findings

Introduction:

line 47 - there "are" a lack of specific training guidelines 

line 56 - Boxing competition, similar to other combat sports competitions, requires both aerobic and anaerobic processes in the effort during a fight. 

lines 66-69 - I might consider describing IHT a little more in terms of what it is to the reader - not everyone reading this journal will understand this method of exercise

line 73 & 75 - superscript 2 for VO2 max

lines 82-83 - similar to above comment regarding elite national, experienced boxers

line 84 - as regards your hypothesis - should you say "will" significantly improve aerobic and anaerobic fitness? 

Methods:

line 89 - see above 

line 99 - I would provide this 230 m in feet also - I think many readers would like this

line 100 - was diet tracked at all?

line 101 - correct degrees Celsius symbol

line 102 - I might say "All boxers fully completed all components of the study." or something to that effect?

Table 1 (and all tables following this) - I might suggest fixing the table so that no words do overlap between lines (e.g., coordination for exercise number 1)

Table 2 - suggestion - would it be better to briefly describe the work to rest ratios and intensities in the text and delete this table? 

line 113 - revise "The studied athletes (national elite boxers) with a...."

lines 117-118 - revise grammar - Begin the sentence with "Basic anthropometric...."

line 122 - "24 hours"

line 126-127 - would you want to put (VO2 max) following maximal oxygen uptake?

line 129 - were all participants male? Perhaps this should be noted somewhere to make it clear?

lines 143-145 & 164-166 - why did you collect blood for a blood lactate measurement 20 min post-exercise? This seems like a really long time post-exercise. 

line 146 & 155 - superscript 2

Did you have any specific criteria for attainment of VO2 max or was it just the "peak" value recored? Perhaps this should be clear in the manuscript as I believe "max" was mentioned previously. 

line 172 - In "the" case...

line 174 - should the Cohen's d be italicized?

line 179 - consider editing beginning of sentence. Perhaps something like "The physical characteristics of boxers..."

line 181 - "significant improvements"

In the statistical analysis section you should include the ranges/values and what they mean for your Cohen's d - e.g., 0.2 = small, 0.5 = medium...

Table 3 - I might add in the word "physical" in the description.

line 202 - "there were" no significant changes in endurance abilities...

In your tables when are the lactate values from? Pre, 3-min post or 20-min post the exercise test? Please clarify.

Discussion:

line 215 - I think it's a bit too much to say "one of the few that has involved high-performance athletes". There is a multitude of studies out there on IHT and athletes in varying sports. 

line 216 - "To the best of our knowledge.."

I like when you readdress the aim in the first discussion paragraph - nice job!!

line 220 - "Both phosphagen and glycolytic systems..." - grammar. Also, how did they improve?

line 225 - grammar

line 226 - please define LL-TH for the reader in this sentence when first mentioned - maybe this should be briefly mentioned in the introduction too?

line 234 - I might say "as seen in the study by... et al. (2020)" for example.

lines 240-245 - consider shortening this sentence.

Try to tie the results of your study in with this paragraph more if possible. 

Again, third discussion paragraph needs to related back to your findings - what were the increases you found in threshold loads in comparison to the studies you mention? If there were no changes then can you perhaps allude to why that might be the case?

line 275 - "thesis" - do you mean hypothesis here?

line 280 - "a.s.l"?

Are there any limitations to your study? These should be noted in a separate paragraph before the conclusions. 

Conclusions - In relation to the great practical implications - I think you must emphasis the point that this training technique may be able to be implemented in elite level settings but is more than likely not possible for the average recreational level boxer. 

Finally, I would like if could get a native English speaker to review the entire manuscript prior to resubmission. 

Author Response

Dear Reviewer,

Thank you very much for your time and valuable comments, which all have been considered and incorporated. The detailed list of responses is given below. We hope that the modifications and explanation will be acceptable for you.

Yours sincerely,

Rydzik, corresponding author

line 22 - would it be correct to say "elite national boxers"?

A: We change it according to suggestion

line 33 - no significant changes "were" noted

A: corrected

line 34 - great practical application but the end of this sentence doesn't really reflect this "great practical application" - I might suggest rewording or adding to the conclusion based on the strength of the findings

A: corrected

line 47 - there "are" a lack of specific training guidelines 

A: corrected

line 56 - Boxing competition, similar to other combat sports competitions, requires both aerobic and anaerobic processes in the effort during a fight. 

A: corrected

lines 66-69 - I might consider describing IHT a little more in terms of what it is to the reader - not everyone reading this journal will understand this method of exercise

A: Due to the editorial limitations the training elements were detailed in tab. 1,2

line 73 & 75 - superscript 2 for VO2 max

A: corrected

lines 82-83 - similar to above comment regarding elite national, experienced boxers

A: corrected

line 84 - as regards your hypothesis - should you say "will" significantly improve aerobic and anaerobic fitness? 

A: corrected

line 89 - see above 

A: corrected

line 99 - I would provide this 230 m in feet also - I think many readers would like this

A: corrected

line 100 - was diet tracked at all?

A: Participants have written their daily meals.

line 101 - correct degrees Celsius symbol

A: corrected

line 102 - I might say "All boxers fully completed all components of the study." or something to that effect?

A: corrected

Table 1 (and all tables following this) - I might suggest fixing the table so that no words do overlap between lines (e.g., coordination for exercise number 1)

A: corrected

Table 2 - suggestion - would it be better to briefly describe the work to rest ratios and intensities in the text and delete this table?

A: According to other reviewers we should leave it

line 113 - revise "The studied athletes (national elite boxers) with a...."

A: corrected

lines 117-118 - revise grammar - Begin the sentence with "Basic anthropometric...."

A: corrected

line 122 - "24 hours"

A: corrected

ine 126-127 - would you want to put (VO2 max) following maximal oxygen uptake?

A: corrected

line 129 - were all participants male? Perhaps this should be noted somewhere to make it clear?

A: corrected

lines 143-145 & 164-166 - why did you collect blood for a blood lactate measurement 20 min post-exercise? This seems like a really long time post-exercise.

A: 20 minutes after the exercise there are clear decreases in lactate level. The difference of the level between LA3 and LA20 can be used to assess regenerative possibilities which could improve due to the training, e.g. because of the improvement of aerobic capacity [1].

  1. Lucertini, F., Gervasi, M., D'Amen, G., Sisti, D., Rocchi, M. B. L., Stocchi, V., & Benelli, P. (2017). Effect of water-based recovery on blood lactate removal after high-intensity exercise. PloS one, 12(9), e0184240.

line 146 & 155 - superscript 2

A: corrected

Did you have any specific criteria for attainment of VO2 max or was it just the "peak" value recored? Perhaps this should be clear in the manuscript as I believe "max" was mentioned previously.

A: VO2max was defined according to the standard criteria of a standardized test.

line 172 - In "the" case...

A: corrected

line 174 - should the Cohen's d be italicized?

A: corrected

line 179 - consider editing beginning of sentence. Perhaps something like "The physical characteristics of boxers..."

A: corrected

line 181 - "significant improvements"

A: corrected

In the statistical analysis section you should include the ranges/values and what they mean for your Cohen's d - e.g., 0.2 = small, 0.5 = medium

A: corrected

Table 3 - I might add in the word "physical" in the description.

A: corrected

line 202 - "there were" no significant changes in endurance abilities

A: corrected

In your tables when are the lactate values from? Pre, 3-min post or 20-min post the exercise test? Please clarify.

A: LA3 is the concentration of lactate 3 minutes after exercise, LA20 – after 20 minutes. LA3-20 is the difference of the speed of metabolizing lactate. It was added in the legend.

line 215 - I think it's a bit too much to say "one of the few that has involved high-performance athletes". There is a multitude of studies out there on IHT and athletes in varying sports. 

A: corrected

line 216 - "To the best of our knowledge.."

A: corrected

line 220 - "Both phosphagen and glycolytic systems..." - grammar. Also, how did they improve?

A: corrected

line 225 - grammar

A: corrected

line 226 - please define LL-TH for the reader in this sentence when first mentioned - maybe this should be briefly mentioned in the introduction too?

A: We added the meaning of this abbreviation

line 234 - I might say "as seen in the study by... et al. (2020)" for example.

A: corrected

lines 240-245 - consider shortening this sentence.

Try to tie the results of your study in with this paragraph more if possible. 

A: corrected

Again, third discussion paragraph needs to related back to your findings - what were the increases you found in threshold loads in comparison to the studies you mention? If there were no changes then can you perhaps allude to why that might be the case?

A: modified

line 275 - "thesis" - do you mean hypothesis here?

A: corrected

linia 280 - „asl”?

A: corrected

Are there any limitations to your study? These should be noted in a separate paragraph before the conclusions.

A: We added limitations

In relation to the great practical implications - I think you must emphasis the point that this training technique may be able to be implemented in elite level settings but is more than likely not possible for the average recreational level boxer. 

A: corrected

Finally, I would like if could get a native English speaker to review the entire manuscript prior to resubmission. 

A: The paper was reviewed by a native speaker

Round 2

Reviewer 3 Report

Nice job on the revisions. Some specific and general comments to help guide the manuscript further: 

The English language should be improved to ensure that an international audience can clearly understand your text. I think you still need to have a native English speaker reread the entire manuscript as there are still too many grammatical errors (e.g.'s, line 33 - "no significant changes in endurance performance was noted after the training" or line 45 - "during a boxing training..." - there are too many of these errors).

Be consistent - e.g., abbreviations - fine tooth comb the edited version and ensure everything is correct (e.g., line 163 - VCO2 - the 2 needs to be a subscript here).

Review discussion to ensure it is related back to the introduction and there is a clear link between both. Consider restating aims at the beginning of the discussion if needed.

I still think you need a better connection from discussion to introduction regarding LL-TH. In my opinion it could be added to the IHT paragraph.

Line 100 - I think it is important to note that the participants' diet was tracked on paper.

Line 103 - add feet conversion from meters.

Specify criteria for attainment of VO2 max.

Ensure it is clear that the sample is male elite level boxers.

Line 303 - should you abbreviate to IHT?

Author Response

Dear Reviewer,

Thank you very much for your time and valuable comments, which all have been considered and incorporated. The detailed list of responses is given below. We hope that the modifications and explanation will be acceptable for you.

Yours sincerely,

Rydzik, corresponding author

The English language should be improved to ensure that an international audience can clearly understand your text. I think you still need to have a native English speaker reread the entire manuscript as there are still too many grammatical errors (e.g.'s, line 33 - "no significant changes in endurance performance was noted after the training" or line 45 - "during a boxing training..." - there are too many of these errors).

A: The manuscript was checked again by the Native Speaker, confirmed by a certificate

Be consistent - e.g., abbreviations - fine tooth comb the edited version and ensure everything is correct (e.g., line 163 - VCO2 - the 2 needs to be a subscript here).

A: All shortcuts have been corrected

Review discussion to ensure it is related back to the introduction and there is a clear link between both. Consider restating aims at the beginning of the discussion if needed.

A: The discussions were checked, the objectives of the work were recalled at the beginning of the discussion

I still think you need a better connection from discussion to introduction regarding LL-TH. In my opinion it could be added to the IHT paragraph.

A: corrected

Line 100 - I think it is important to note that the participants' diet was tracked on paper.

A: added to text

Line 103 - add feet conversion from meters.

A: The training was conducted at an altitude measured in meters above sea level. At the request of the reviewer, conversion into rates is given in parentheses (verse 109). Foot = 0.3048m

Specify criteria for attainment of VO2 max.

A: The criteria for determining VO2max were as follows: (1) plateau in oxygen consumption, (2) a respiratory exchange ratio of>1.15 and (3) attainment of a heart rate within 10 beats per minute of age-predicted maximum (Howley, Bassett, & Welch, 1995).. According to paper by Edvardsen et. al. (Edvardsen, E., Hem, E., & Anderssen, S. A. (2014). End criteria for reaching maximal oxygen uptake must be strict and adjusted to sex and age: a cross-sectional study. PloS one, 9(1), e85276) - (sample of 861) only forty-two percent of the participants achieved a plateau in VO2. Our observations were similar - in some cases no plateau in VO2 was observed; If no plateau was observed but the rest of the criteria are met, VO2peak can be assumed as VO2max (Howley, Bassett, & Welch, 1995), but for accuracy we decided to use in our paper term ‘VO2peak’ instead of ‘VO2max’. VO2peak is defined in methods section.

Ensure it is clear that the sample is male elite level boxers.

A: The information is described in the text. The research was conducted on elite male boxers

Line 303 - should you abbreviate to IHT?

A: corrected
